# Tungsten and Copper (II) Oxide Mixtures as Gasless Time Delay Compositions for Mining Detonators

**DOI:** 10.3390/ma16103797

**Published:** 2023-05-17

**Authors:** Marcin Gerlich, Marcin Hara, Waldemar A. Trzciński

**Affiliations:** 1Faculty of Advance Technology and Chemistry, Military University of Technology, Kaliskiego 2, 00-908 Warsaw, Poland; marcin.gerlich@wat.edu.pl (M.G.); marcin.hara@wat.edu.pl (M.H.); 2NITROERG S.A., Alfred Nobel Square 1, 43-150 Bierun, Poland

**Keywords:** time delay compositions, detonators, solid state reactions, burning rate

## Abstract

The widespread use of pyrotechnic compositions in time delay detonators is the reason for research aimed at expanding knowledge of the combustion properties of new pyrotechnic mixtures, whose components react with each other in the solid or liquid state. Such a method of combustion would make the rate of combustion independent of the pressure inside the detonator. This paper presents the effect of the parameters of W/CuO mixtures on their properties of combustion. As this composition has not been the subject of previous research and is not described in the literature, the basic parameters, such as the burning rate and the heat of combustion, were determined. In order to determine the reaction mechanism, a thermal analysis was performed, and the combustion products were determined using the XRD technique. Depending on the quantitative composition and density of the mixture, the burning rates were between 4.1–6.0 mm/s and the heat of combustion in the range of 475–835 J/g was measured. The gas-free combustion mode of the chosen mixture was proved using DTA and XRD. Determination of the qualitative composition of the combustion products and the heat of combustion allowed estimation of the adiabatic combustion temperature.

## 1. Introduction

It is estimated that only 20–30% of the energy released in the process of detonation is used for deposit fragmentation. The rest of the energy is lost through the generation of paraseismic vibrations, heat and noise [1]. Therefore, it is in our interest to find and apply proper methods for increasing blasting performance. Due to both safety and economic issues, a number of studies are being carried out to optimize blasting operations in order to minimize paraseismic vibrations [2,3,4]. These studies have shown that the accuracy of the sequential firing of explosive charges is an important aspect of blasting works. For this reason, numerous studies have been conducted to develop accurate time delay compositions for detonators.

It is indicated that the pressure inside the detonator significantly affects the accuracy of the currently used time delay compositions. Many pyrotechnic mixtures, during the combustion of which the main exothermic reaction takes place at the solid–gas or liquid–gas interface, are characterized by a variable burning rate, which is a function of the length of the delay system [5,6]. Therefore, it is recommended to use mixtures whose components react with each other in the condensed state. Such pyrotechnic mixtures are called gasless mixtures. Of course, due to the use of organic binders or a system of several oxidants, at least one of which can decompose at a temperature lower than the reaction temperature in the solid state, the term is also used for mixtures generating less than 10 mL of gaseous products per 1 g of mixture [7]. Several pyrotechnic mixtures have been proposed in recent years that meet these conditions (e.g., Ti/C-3Ni/Al [8], Fe/BaO_2_ [9], Sb/KMnO_4_ [10]). Unfortunately, the high ignition temperature and high heat losses due to the dissipation of energy through the walls of the time delay element make many gasless mixtures impractical due to the possibility of stopping the combustion wave. The vast majority of mixtures currently being developed fall into the so-called low-gas pyrotechnic mixtures giving off little more than 10 mL of gaseous products per gram of mixture (e.g., B/BaCrO_4_ [11]). Therefore, research is needed to develop stable-burning gasless compositions.

In the work [12], the burning rates of Al, B and Zr metal powders mixed with metal oxides were calculated. The resulting burning rates ranged from 21 to 600 mm s^−1^. It is expected that the use of tungsten in retarding mixtures will significantly reduce these rates. In the paper [13], tungsten powders of different fineness were studied in mixtures with KClO_4_ and BaCrO_4_. Very low velocities of the order of 0.6 mm s^−1^ were obtained, and a lot of unsuccessful attempts were found (from 5.8 to 25.5 percent). The results of a comprehensive study of W/MnO_2_ mixtures (%W = 40, 50, 60 and 70 wt.%) are presented in the paper [14]. Measured maximum combustion temperatures ranged from 1466 to 1670 K, depending on the composition of the tested mixture. The measured burning rates were from 1.62 to 4.61 mm s^−1^. The maximum measured gas production was 9.1 mL g^−1^. The composition of the solid combustion products was also determined with powder X-ray diffraction.

In this paper, the results of research on a gasless W/CuO mixture are presented. First, the basic parameters, such as the burning rate and the heat of combustion, were determined. Then, in order to determine the reaction mechanism, a thermal analysis was performed and the combustion products were determined using the XRD method. We expect the stable burning rates of the new pyrotechnic mixtures to be on the order of several mm s^−1^ and to find application in decisecond time delay detonators.

## 2. Materials and Methods

### 2.1. Materials

Micron-sized tungsten and copper (II) oxide were used. Purity of both powders was 99%. The grain size analysis was performed using ANALYSETTE 22 MicroTec plus laser particle size analyzer manufactured by Fritsch GmbH, Idar-Oberstein, Germany.

For the preparations of time delay elements, tubes made from an alloy of zinc and aluminum (ZnAl) were used. The tube length was 26 mm with inner diameter of 3 mm and outer diameter of 6 mm. ZnAl tubes were prepared by casting. All tubes were checked for proper diameters and cleaned from the machine oil that covers them after casting. Cleaned tubes were covered by thin layer of graphite to prevent unnecessary friction during the elaboration of the pyrotechnic mixture.

### 2.2. Sample Preparation

W/CuO mixtures with tungsten content from 20 wt.% to 70 wt.% were prepared. About 100 g of each mixture was prepared using a TURBULA^®^ dry mixer. For every sample, the mixing time was 6 h. As the components of the composition are not hygroscopic, prior drying of the components was not necessary.

The time delay elements preparation was based on the multi-stage pressing of small doses of W/CuO composition. The doses were applied by volume. It was found that this method does not cause differences in dose weight greater than 2 wt.%. It does not affect the height of the sample to the extent that it could cause noticeable differences in its density after compression. The elaboration was performed using a hydraulic press with the strain gauge that indicated elaboration force. Samples were prepared under 200, 300, 400 and 500 MPa pressure. The number of doses depended on both the pressing pressure and the quantitative composition of the mixture (the smallest number of doses was 8, and the largest was 12).

In order to control the accuracy of elaboration, every ZnAl tube was weighed before and after filling with a time delay mixture. Knowing the mass of pressed mixture and the dimensions of the ZnAl tube (volume of the mixture), the density of the mixture as a function of tungsten content and elaboration pressure was determined.

### 2.3. Sensitivity to Impact and Friction

In order to determine the sensitivity of W/CuO compositions to impact, the BAM apparatus with a 5 kg hammer was used in accordance with the method given in the standard [15]. The highest energy at which no initiation is observed in six consecutive trials was assumed as the sensitivity to impact. The sensitivity to friction was determined on a Peters apparatus in accordance with the method described in the standard [16]. The greatest friction force at which no initiation was observed in six consecutive trials was assumed as the sensitivity to friction.

### 2.4. Burnig Rate Measuring

The rate of the combustion front propagation was measured by determining the delay time of electric detonators containing time delay elements with the investigated compositions inside them. The scheme of an electric detonator is presented in Figure 1. With the known length of pressed pyrotechnic mixture and the time of combustion, the mean burn rate was calculated.

Copper wires and aluminum shells were applied to the detonator. Primary charge was lead azide (Pb(N_3_)_2_), while the secondary charge was pressed penthrite (PETN). The energy of the pyrotechnic fuse head was about 0.12 kJ (this parameter was calculated theoretically).

The delay time of electric detonators was measured using an ohmmeter-microphone system. Time measurement was started by the computer after applying the appropriate current pulse and ended after receiving a signal from the microphone indicating the detonation of the secondary charge. The time measurement was performed with an accuracy of 0.1 ms. The microphone was placed 2 m from the detonator. Therefore, the obtained results were corrected for the time needed for the shock wave to travel this distance. In order to obtain results with the highest possible accuracy, the average fuse head delay time was also subtracted from the total delay time of the detonator.

### 2.5. Differential Thermal Analysis

Thermal analysis of the W/CuO mixture containing 50 wt.% of tungsten was performed using Labsys thermal analyzer manufactured by Setaram Instrumentation. In order to test the sample in an inert gas atmosphere, the analyzer was extended with a Bronkhorst mass flow controller. It allowed us to set the exact inert gas flow. The analysis consisted in placing about 3 mg of the sample in a crucible made of Al_2_O_3_ with a capacity of 100 µL. The argon flow was about 50 mL min^−1^. The sample was heated at 10 °C min^−1^ up to 1100 °C.

### 2.6. Heat of Combustion

The combustion heat was measured for dry-mixed samples of about 10 g. Each sample was placed in a 4 mL quartz crucible. The KL-12 calorimeter was used for measurements. The calorimeter consists of a 350 mL steel bomb placed inside a steel vessel with 2750 mL of distilled water. A thin resistance wire made from Kanthal D alloy (69% Fe, 22% Cr, 6% Al, 3% Zn and Si) was used for the ignition of W/CuO mixtures. All tests were carried out in an argon atmosphere under a pressure of 2 MPa. To avoid oxygen contamination, a standard procedure of purging the contents of the calorimetric bomb with argon for a period of 120 s was applied.

### 2.7. Combustion Products Analysis

The qualitative composition of solid combustion products was determined via XRD method using the Rigaku SmartLab 3 kW diffractometer 124 with the Cu lamp (40 kV, 30 mA) and the linear detector 1D: Dtex250. The results were analyzed using the ICDD PDF4 + 2022 databases and PDXL analytical program. Measurement was performed in Bragg–Brentano (BB) theta-2theta geometry in the 2θ range of 10°–90° with the step of 0.02° and the traverse speed of 2° min^−1^. Powder samples were placed on a “backgroundless” table, which was Si510 single crystals. Phase composition was determined automatically using XRD library (ICDD-PDf4-2020) and PDXL program.

## 3. Results and Discussion

### 3.1. Particle Size Analysis

One of the most important parameters of retardant mixture components is their grain characteristics. It is indicated that the decrease in the average particle diameter increases the burning rate due to the increase in the contact surface of the components and the elimination of the diffusion hindrance. Figure 2 and Figure 3 show the grain characteristics of the components used during this research.

Based on the measurement results, the volume-weighed mean diameter *D*[4,3] and the diameter D99 for tungsten and copper (II) oxide grains were determined (Table 1).

### 3.2. Sensitivity

Sensitivity testing of tungsten and copper oxide mixtures showed no sensitivity to friction or impact. In the friction sensitivity test, no signs of ignition were registered after applying a friction force of 360 N (in six repeated tests), while in the case of the impact sensitivity test, dropping a 10 kg hammer from a height of 50 cm (after applying an energy of 50 J) did not cause visible signs of ignition. Therefore, the mixtures tested are safe.

### 3.3. Burning Rate

In many cases, literature on pyrotechnic time delay compositions is silent about the loading conditions. It should be remembered that, apart from the qualitative composition, a parameter such as the loading pressure is one of the most important factors in the technology of time delay detonators. However, the value of the loading pressure alone does not provide sufficient information on the state of the pyrotechnic mixture. Mainly due to the fact that the final density of the mixture, in this case, also depends on the size of a single, pressed dose. In the industry, it is commonly known that differences in the time delay composition density can occur when using too large a volume of a single dose. This is because the pressing creates force vectors also perpendicular to the axis of the sample. Therefore the pressure force decreases with the height of the dose due to friction. Thus, the smallest possible portions of time delay mixtures should be used. The dependence of the mixture density on the loading pressure is presented in Figure 4. The figure also shows the mean absolute deviation for the measured densities.

The relationships shown in Figure 4 show that the density of any W/CuO mixture increases almost linearly with loading pressure. The change in the density of the mixtures studied relative to their maximum theoretical density (MTD) was compared. The MTD was calculated by assuming a tungsten density of 19.3 g cm^−3^ and a CuO density of 6.3 g cm^−3^. It turned out that for a loading pressure of 200 MPa, the relative density of the mixtures varies slightly from 80 to 81.5% MTD, while for a pressure of 500 MPa, the relative density decreases from 90% MTD (20 wt.% of W) to 86% MTD (70 wt.% of W). For high-loading pressures, copper (II) oxides are easier to press than tungsten powder.

The delay compositions with 40–70 wt.% tungsten content were tested in burn rate measurements. The mixtures with a tungsten content under 40 wt.% were very hard to initiate and did not sustain a steady combustion for any applied pressing load. The average porosity of the four mixtures with tungsten contents ranging from 40 to 70 wt.%, calculated from actual and theoretical densities, decreased with increasing the loading pressure and was 19.4 ± 0.7%, 16.4 ± 0.8%, 13.4 ± 0.9% and 11.3 ± 1.2% for the 200, 300, 400 and 500 MPa pressures, respectively. However, porosity did not change monotonically with tungsten content for a given loading pressure. Hence the deviation from the average value is also given. Figure 5 presents the results of the W/CuO composition burn rate tests as a function of tungsten content and loading pressure. In general, mixtures with higher porosity burn faster.

It was found that mixtures of tungsten and copper (II) oxide loaded in ZnAl tubes burn at a rate of about 4.1 to 6.0 mm s^−1^, reaching a maximum loading pressure of 60 wt.% tungsten content. It is clearly visible that the burning rate drops for the W/CuO mixtures with a tungsten content above 60 wt.%. It is suspected that the main reason for this is the high thermal conductivity of tungsten (183 W m^−1^ K^−1^ at 280 K [17]), which may be present in these mixtures due to their potentially negative oxygen balance. Together with the relatively high thermal conductivity of the time delay element ZnAl tubes, the heat generated by combustion is quickly collected and transferred both ahead of the combustion front (which should not significantly affect the burning rate as it heats the substrates) and towards the combustion products.

### 3.4. Thermal Analysis

In order to determine the mechanism of the combustion reaction of the tested mixtures, thermal analysis was used to identify the temperature and phase type of the components when the main exothermic reaction took place. In the case of gasless compositions, the exothermic effect is expected to occur at a temperature lower than the temperature of thermal decomposition of the oxidant (here, CuO, for which this temperature varies from 900 to 1050 °C [18,19]). Figure 6 shows a thermogram of a W/CuO mixture containing 50 wt.% of tungsten. Two effects are visible on the DTA curve—the first large exothermic effect starting at about 606 °C (combustion of the mixture) and the second, much weaker effect starting at about 1065 °C (thermal decomposition of trace amounts of unreacted CuO). The analysis of the TG curve indicates the lack of gaseous combustion products that would cause a decrease in the mass of the sample after the initiation of combustion at about 606 °C.

The above analysis indicates that tungsten reacts with copper oxide in the condensed phase (at the solid–solid interface), and the W/CuO pyrotechnic mixture is actually a gasless mixture.

### 3.5. Heat of Combustion

To better understand the reaction between tungsten and copper (II) oxide, the heat of combustion was determined both empirically and theoretically. Two calorimetric measurements were made for each mixture. Measurements were additionally made for the mixture containing 45 wt.% of tungsten. Figure 7 shows the dependence of the heat of combustion on tungsten content. The figure also shows the deviation from the average heat value. The shape of the dependence indicates that stoichiometric proportions of the components are to be expected for compositions with a tungsten content of less than 50 wt. %. The highest heat of combustion was obtained for samples with a fuel content of 45 wt.%.

Copper particles were visible in all samples after burning in the calorimetric bomb. The hardness and compressibility of the samples indicate that some of the combustion products were in the liquid state. For example, Figure 8 shows samples of the mixture with 50 wt.% of W before and after heat measurement.

The measured heats of combustion were compared with those calculated for the assumed tungsten oxides in the combustion products. The enthalpy of the reaction was calculated using Hess’s law:(1)$$\Delta H_{r}=\overset{k}{\underset{i=1}{\sum }}n_{i}{\left(\Delta H_{fT_{0}}^{0}\right)}_{i}-\overset{l}{\underset{j=1}{\sum }}n_{j}{\left(\Delta H_{fT_{0}}^{0}\right)}_{j},$$
where ${\left(\Delta H_{fT0}^{0}\right)}_{i}$ is the enthalpy of production of the *i*-th combustion product at the standard condition (*p*_0_ = 1 atm, *T*_0_ = 298.15 K), *n_i_*—the number of moles of the *i*-th combustion product, *k*—the number of combustion products, ${\left(\Delta H_{fT0}^{0}\right)}_{j}$—the enthalpy of formation of the *j*-th substrate of the composition, *n_j_*—moles of *j* component and *l*—number of pyrotechnic components. The values of enthalpies of formation for the components of the mixtures studied and their combustion products were taken from the JANAF tables [20].

For further consideration, the following reactions were assumed as most possible:(2)$$W+2\;\mathrm{CuO}\;\to {\;\mathrm{WO}}_{2}+2\;\mathrm{Cu}\;\;\;\;\;\;\Delta Hr=-809.4\;J\;g^{-1}$$
(3)$$W+2.72\;\mathrm{CuO}\;\to {\;\mathrm{WO}}_{2.72}+2.72\;\mathrm{Cu}\;\;\;\Delta Hr=-891.2\;J\;g^{-1}$$
(4)$$W+2.9\;\mathrm{CuO}\;\to {\;\mathrm{WO}}_{2.9}+2.9\;\mathrm{Cu}\;\;\;\;\Delta Hr=-886.5\;J\;g^{-1}$$
(5)$$W+2.96\;\mathrm{CuO}\;\to {\;\mathrm{WO}}_{2.96}+2.96\;\mathrm{Cu}\;\;\;\Delta Hr=-889.6\;J\;g^{-1}$$
(6)$$W+3\;\mathrm{CuO}\;\to {\;\mathrm{WO}}_{3}+3\;\mathrm{Cu}\;\;\;\;\;\Delta Hr=-887.0\;J\;g^{-1}$$

The level of tungsten oxidation has an impact on the stoichiometric amount of fuel, which for the above reaction is 53.6%; 45.9%; 44.3%; 43.8% and 43.5%, respectively.

Experimental heat of combustion values was compared with theoretical values calculated for different tungsten contents assuming one of the reactions (2)–(6). In the case of an excess of copper (II) oxide in the mixture, it was assumed that the product was also CuO. The results of the theoretical estimation of the heat of reaction (Q*_cal_* = −Δ*H_r_*) are presented in Figure 9. From the heat of reaction point of view, there is no significant difference between reactions forming WO_2.72_, WO_2.9_, WO_2.96_ and WO_3_. It should be noted that the experimental values of the heat of combustion correlate well with those calculated assuming the presence of tungsten oxides, in which there are 2.72 to 3 O atoms per W atom. A clear difference, however, is evident in the case of WO_2_.

### 3.6. Combustion Products Analysis

The residue in the calorimetric bomb was subjected to XRD analysis. An example diffractogram obtained for the combustion products of W/CuO (%W = 60 wt.%) is shown in Figure 10. The results of the analysis are shown in Table 2. The estimated weight fractions of individual solid products in the residue are also included. Unfortunately, the weight contribution of individual elements in the residue does not correlate with their contribution to the starting mixture.

The table shows the lack of tendencies in the occurrence of specific tungsten compounds depending on the oxygen balance of the mixture from which they were obtained. For example, WO_2_ oxide appears as expected for mixtures with negative oxygen balance, as does unreacted tungsten. However, the lack of reflections characteristic of copper in the case of a mixture containing 40% of tungsten is puzzling. In addition, it was noted that if Cu_2_WO_4_ was detected in the combustion products of the mixtures, W_18_O_49_ was not.

The way to identify errors in the XRD analysis, e.g., due to the occurrence of some compounds in the amorphous phase, is to compare the obtained diffractogram with the theoretical diffraction pattern that would arise if only those compounds identified by the database were present in the sample. For example, Figure 11 shows the comparison of the diffractogram obtained for the products of W/CuO (%W = 40 wt.%) with the calculated pattern, including identified compounds (Cu_2_WO_4_, W_8_O_21_, WO_2.96_, WO_3_). For comparison, the reflections obtained for Cu in the products of W/CuO (%W = 50 wt.%) composition are also included in Figure 11.

The differences in the intensity of reflections present in the diffractogram obtained experimentally and in the diffraction pattern determined theoretically can be neglected, as they result from the differences between the actual amount of a given compound in the sample and the amount determined in the XRD test. The essential fragments of the diffractogram determined theoretically are the fragments on which the reflections present in the diffraction pattern do not appear. This situation occurs at 2θ angles of approximately 43°, 50° and 73°. As can be seen from the fragment with the locations of reflections for copper (added at the bottom of Figure 11), it can be suspected that not all components of the mixture have been well identified, as these values coincide. This means that, in this case, metallic copper should also be added to the combustion products of the W/CuO mixture. This seems obvious from the stoichiometry of the reaction. It is impossible to synthesize Cu_2_WO_4_ without precipitating copper, as the number of oxygen atoms in tungstate is twice as high as that of copper, so the synthesis reaction must be as follows:(7)$$W+4\;\mathrm{CuO}\;\to {\;\mathrm{Cu}}_{2}{\mathrm{WO}}_{4}+2\;\mathrm{Cu}\;\;\;\Delta Hr=-883.52\;J\;g^{-1}$$

Therefore, assuming that copper (I) tungstate (VI) will be the only product of tungsten oxidation, two copper atoms should be produced for each tungstate molecule, which would be detected during XRD. However, it should also be noted that copper (I) tungstate (VI) is, in fact, not the only combustion product of a mixture containing 40 wt.% tungsten—its oxides such as W_8_O_21_, WO_2.96_, WO_3_ are also present, therefore it must be assumed that the lack of copper in the results is due to difficulties in identifying the qualitative composition of a system with this structure.

Additionally, microscopic photographs of combustion products were taken using an optical microscope AXIO Lab.A1 manufactured by Carl Zeiss, Jena, Germany. Figure 12 shows the structure of combustion products obtained from the tested mixtures (i.e., containing 40–70 wt.% tungsten). All images except (b) show the structure of the products with ×10 magnification. Image (b) is a 40× zoom. Figure 12 indicates that the tungsten oxides (WO_x_) obtained from combustion are characterized by a highly developed surface. In each case, they form microrods extending from spherical WO_x_ aggregates. This crystalline structure is most often associated with W_18_O_49_ [21]. The violet-blue color in Figure 12a,b also corresponds to W_18_O_49_ [22]. However, the analysis of Table 2 indicates the absence of this oxide in the combustion products of mixtures containing 40 and 60 wt.% of tungsten. Nevertheless, the literature indicates that tungsten (VI) oxide can form nanowire-shaped crystals under certain conditions [23].

Figure 12c shows the combustion products of a mixture containing 50 wt.% tungsten. It shows that the produced oxides are characterized by the densest microfibrous structure. The results of the XRD tests presented in Table 2 indicate the presence of W_18_O_49_ oxide in the structure of the products, which according to the literature, is characterized by the most fibrous structure. A similar qualitative composition of the products is shown in Figure 12d; however, due to the strongly negative oxygen balance of the mixture, the amount of produced oxides is lower, which translates into the density of the structure. It is worth noting that the densest structure of the combustion products of the W/CuO mixture was obtained for the quantitative composition close to the stoichiometric one calculated on the basis of reaction (3), which is about 46 wt.% W. This is also consistent with the results of calorimetric tests, which confirmed that the largest amount of energy (about 840 J g^−1^) is generated for a mixture containing 45 wt.% W.

Figure 12e shows the combustion products of the mixture with the most negative oxygen balance. Due to the significant oxygen deficiency in the system, a small amount of tungsten oxides is observed. In this case, a microrod structure is also obtained.

The authors do not draw conclusions about the composition of the combustion products based on the color of the samples, as it also depends on the intensity with which the sample was irradiated during the taking of microscopic images. Moreover, examples of chromism of WO_x_ oxides are very well documented in the literature [24,25]. For this reason, the presence of a given oxide cannot be inferred from the color shown in Figure 12 alone, as the high temperature at which these compounds were synthesized and the possible presence of copper cations in the defective WOx structure [26] can cause significant color changes.

### 3.7. Adiabatic Temperature of Combustion

The measured heat of combustion and composition assumptions of the combustion products allow the calculation of the adiabatic temperature of combustion. First, a polynomial regression describing the temperature dependence of the specific heat for the components of CuO/W mixtures and their combustion products was carried out using data from the JANAF tables [20], which give the specific heat of the compounds over a wide temperature range:(8)$$C_{p}=C_{1}+C_{2}\theta +C_{3}\theta^{2}+C_{4}\theta^{3}+C_{5}\theta^{-2},$$
where *C*_1_, *C*_2_,…, *C*_5_ are constant coefficients, *θ* = *T*/1000 and *T*—absolute temperature. For the simpler *C_p_*(*T*) relationship, a linear function was used.

These functions were determined for each compound assumed to be present in the combustion products. This made it possible to determine the energy that must be added to the system to heat the combustion products of the W/CuO mixture to a given temperature *T_a_* and to compare it with the heat determined experimentally:(9)$$Q_{exp}=\overset{n}{\underset{i=1}{\sum }}x_{i}\left\{\overset{T_{m,i}}{\underset{T_{0}}{\int }}{\left[C_{ps}\left(T\right)\right]}_{i}dT+{\left(\Delta H_{m}\right)}_{i}+\overset{T_{a}}{\underset{T_{m,i}}{\int }}{\left[C_{pl}\left(T\right)\right]}_{i}dT\right\}\;,\;$$
where ${\left[C_{p}\left(T\right)\right]}_{i}$ is the specific heat of the *i*-th combustion product at the temperature *T* (the subscript *s* refers to solid and *l* to liquid), *x_i_* is the mole fraction of the *i*-th product, *T_m,i_* and (Δ*H_m_*)*_i_* are the melting temperature and the melting enthalpy of the *i*-th product, respectively, *n*—the number of combustion products and *T*_0_ = 298.15 K.

Equation (9) can be used to determine the adiabatic combustion temperature, *T_a_*. In this way, the temperatures of combustion of the tested mixtures W/CuO were determined assuming specific compositions of combustion products. One of the tungsten oxides listed in reactions (2)–(6) was taken as the main combustion product.

The results of the calculations are presented in Figure 13. Due to the apparent discrepancy between the measured heat and the calculated heat assuming the presence of only WO_2_ in the products, an estimation of the adiabatic combustion temperature for this case was abandoned. It was assumed that copper melts in a combustion wave at 1385 K, and its enthalpy of fusion is 13,138 kJ mol^−1^ [20]. Since the JANAF tables for WO_2.72_, WO_2.90_ and WO_2.96_ provide heat capacity values only for the solid phase, the combustion temperature was also estimated for solid WO_3_. For comparison, the adiabatic combustion temperature was also calculated, assuming that WO_3_ melts at 1745 K with an enthalpy of fusion of 73.429 kJ mol^−1^ [20]. It turned out that the calculated adiabatic combustion temperatures for all tungsten contents were lower than the melting point of WO_3_. Thus, Figure 13 shows the calculations performed assuming that 50% of WO_3_ is melted in the combustion zone. The analysis of Figure 13 shows that among the four tested mixtures, the mixture containing 45 wt.% of tungsten has the highest combustion temperature. This temperature is 2000 K assuming solid WO_3_ in the combustion products. The temperatures of mixtures with WO_2.72_, WO_2.9_ or WO_2.96_ combustion products are close to each other. Taking the WO_3_ melting into account definitely reduces the calculated combustion temperatures. In the actual process, the combustion temperatures should be within the estimated range.

## 4. Conclusions

This paper presents the results of the study of the burning rate of the W/CuO mixture as a function of tungsten content and reloading pressure. It has been shown that burning rates from 4.1 to 6.0 mm s^−1^ are for mixtures containing tungsten in the range of 40–70%. The low dependence of the burning rate on the fuel content is prospective in terms of the application of tested mixtures. In addition, an increase in the loading pressure from 200 to 500 MPa, associated with decreasing porosity, causes a decrease in the burning rate by only about 0.5 mm s^−1^, which is also a positive aspect from the technological point of view.

Thermal analysis showed that the W/CuO mixture with 50 wt.% tungsten is a gasless mixture during combustion in which the components react with each other in the solid state. This is indicated by the fact that the ignition temperature of the mixture is lower than the thermal decomposition temperature of the CuO oxidant. The flash point of the tested mixture is about 880 K.

The main combustion products of the mixture are WO_2_, WO_2.72_, WO_2.96_, WO_3_, Cu_2_WO_4_, metallic copper and tungsten. For mixtures with a high combustion temperature, WO_2_ is not observed due to the disproportionation reaction to tungsten and WO_3_. The heat of combustion of the tested mixtures ranges from about 475 to 835 J g^−1^. The maximum amount of energy per gram of the mixture is obtained for a mixture with a tungsten content equal to 45% by mass. Calculations of the adiabatic combustion temperature (based on the heat of combustion) showed that the highest temperature of 2000 K was obtained, assuming only solid WO_3_ formation in the mixture containing 45% tungsten. If one from other solid tungsten oxides was assumed in the combustion products, the temperature was slightly lower (1075 K). The lowest combustion temperature from 1700 K to 1820 K was obtained for mixtures containing 70% tungsten (i.e., with a strongly negative oxygen balance). The assumption that 50% of WO_3_ is melted in the flame zone definitely lowered the maximum calculated combustion temperature from 2000 K to 1830 K.

The research showed that the W/CuO mixtures are characterized by stable combustion at burning rates characteristic of decisecond pyrotechnic mixtures. Furthermore, due to the non-hygroscopic nature of the components, they are insensitive to the humidity prevailing during their elaboration. These features indicate that they can be considered promising and used in the production of time delay detonators.

## Figures and Tables

**Figure 1 materials-16-03797-f001:**
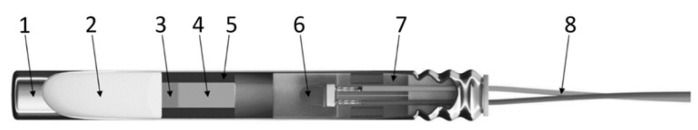
Cross-section of an electric detonator. 1—shell, 2—secondary charge, 3—primary charge, 4—time delay composition, 5—ZnAl tube, 6—fuse head, 7—shielding plug, 8—wire.

**Figure 2 materials-16-03797-f002:**
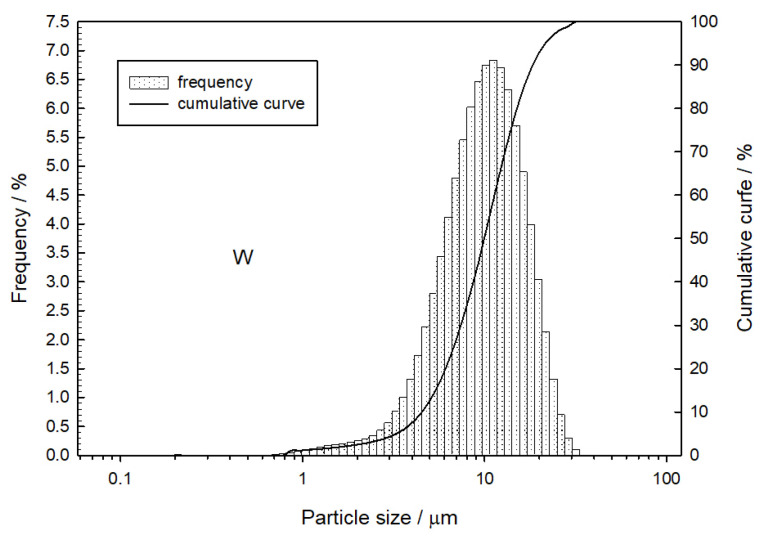
Particle size distribution of tungsten.

**Figure 3 materials-16-03797-f003:**
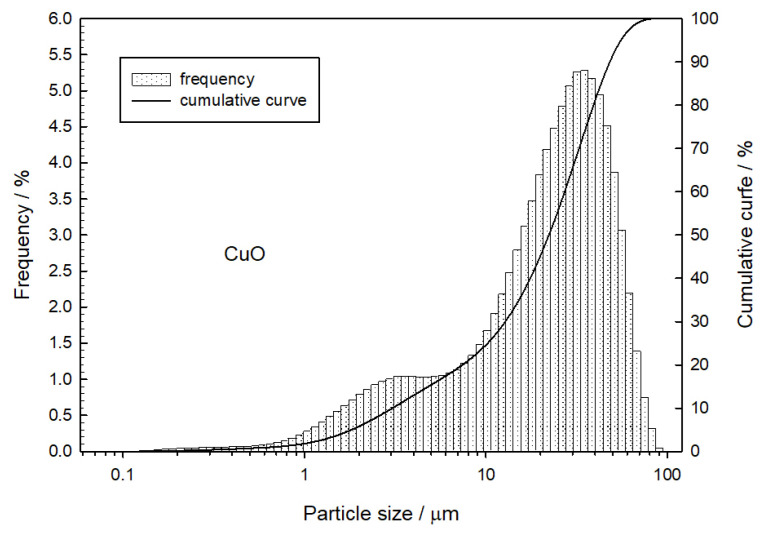
Particle size distribution of copper (II) oxide.

**Figure 4 materials-16-03797-f004:**
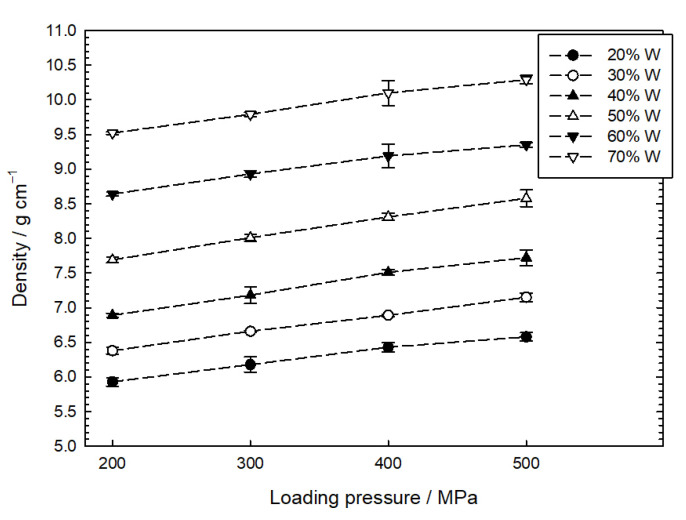
The density of time delay mixture depends on the loading pressure for different tungsten content.

**Figure 5 materials-16-03797-f005:**
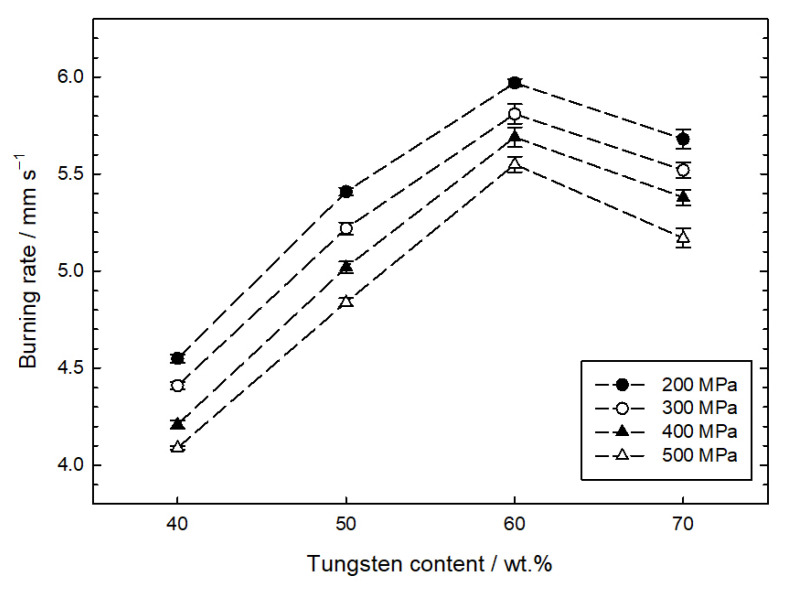
Influence of loading pressure on the dependence of burning rate on tungsten content.

**Figure 6 materials-16-03797-f006:**
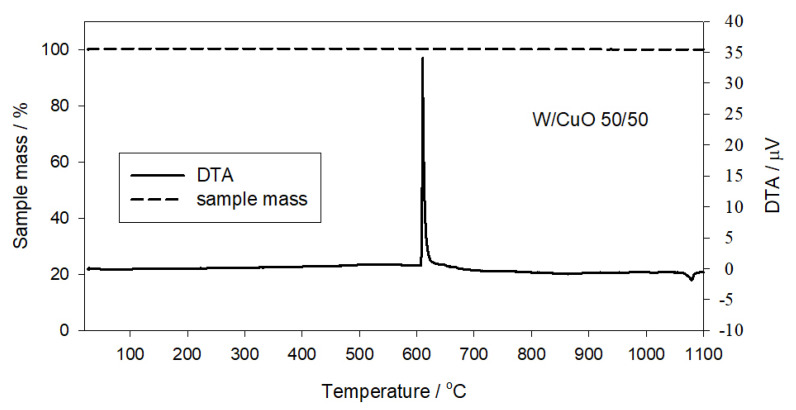
Thermogram of W/CuO mixture with 50 wt.% tungsten (temperature rate 10 K min^−1^, argon atmosphere).

**Figure 7 materials-16-03797-f007:**
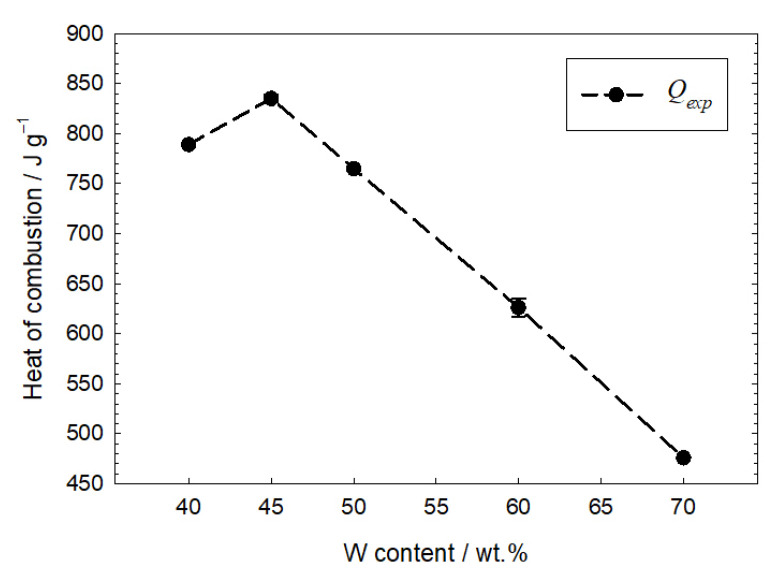
Calorimetric heat of combustion as a function of W content.

**Figure 8 materials-16-03797-f008:**
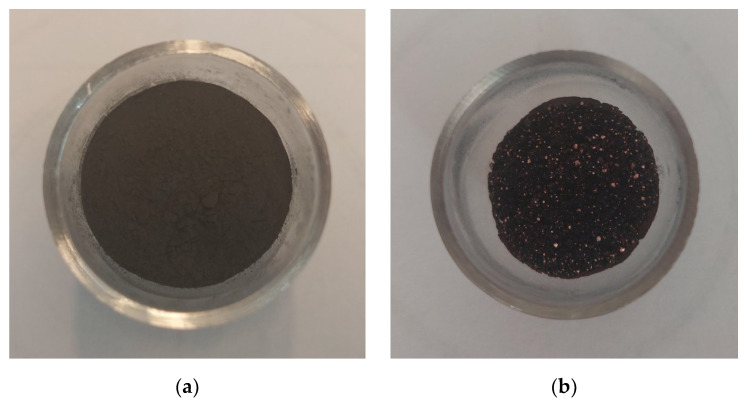
Samples of mixture with 50 wt.% of W in a quartz crucible before (**a**) and after (**b**) heat measurement.

**Figure 9 materials-16-03797-f009:**
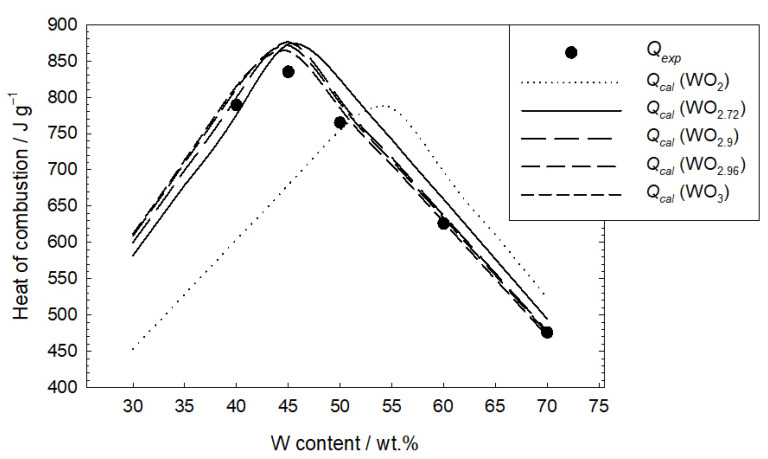
Comparison of the experimental heat of combustion of the tested mixtures with the heat estimated theoretically for the assumed tungsten oxides in the combustion products.

**Figure 10 materials-16-03797-f010:**
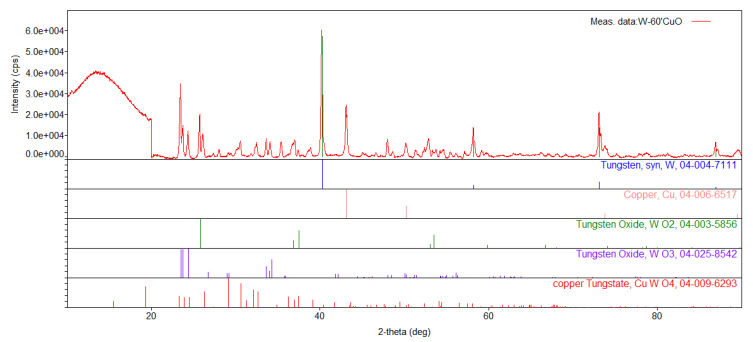
Diffractogram of the combustion products of the W/CuO mixture containing 60 wt.% of tungsten.

**Figure 11 materials-16-03797-f011:**
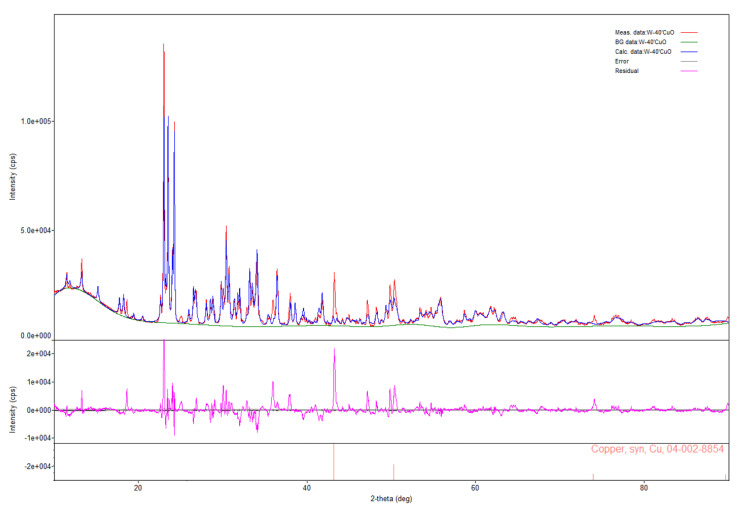
Comparison of the diffractogram obtained for the W/CuO combustion products (%W = 40 wt.%) with the curve determined theoretically for the identified products.

**Figure 12 materials-16-03797-f012:**
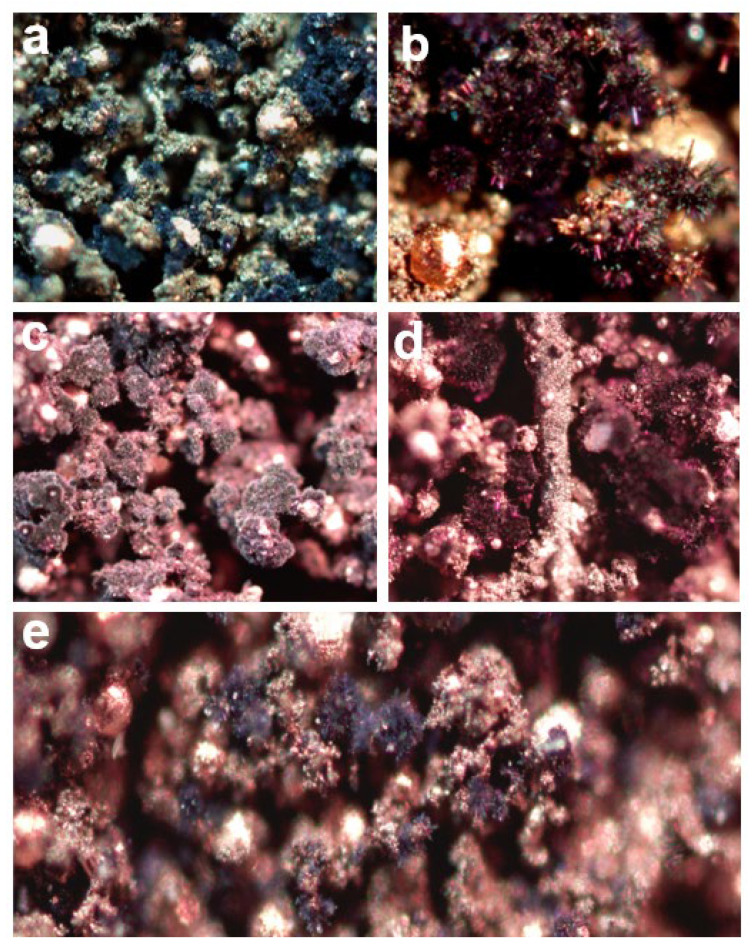
Optical microscope images of W/CuO combustion products for mixture containing 40 wt.% W (**a**,**b**), 50 wt.% W (**c**), 60 wt.% W (**d**) and 70 wt.% W (**e**).

**Figure 13 materials-16-03797-f013:**
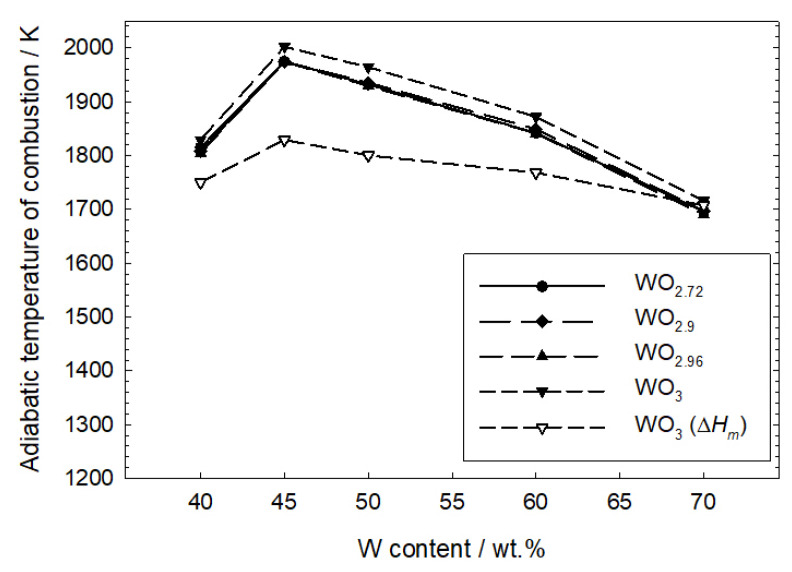
Adiabatic combustion temperatures are estimated from the experimental heat of combustion of the tested W/CuO mixtures.

**Table 1 materials-16-03797-t001:** Grain characteristics for tungsten and copper (II) oxide particles.

Powder	*D*[4,3] [µm]	*D*_99_ [µm]
Tungsten	10.5	25.6
Copper (II) oxide	24.3	68.8

**Table 2 materials-16-03797-t002:** Solid combustion products identified by XRD technique from calorimetric bomb residues.

Content of W [wt.%]	Combustion Products (wt.%)							
40	Cu_2_WO_4_ (51.3)		W_8_O_21_ (9.1)		WO_2.96_ (8.1)	WO_3_ (31.5)		
50				W_18_O_49_ (WO_2.72_) (70.2)				Cu (29.8)
60	Cu_2_WO_4_ (26.1)	WO_2_ (8.3)				WO_3_ (16.5)	W (22.9)	Cu (26.2)
70		WO_2_ (14.7)		W_18_O_49_ (WO_2.72_) (31.9)			W (28.6)	Cu (24.8)

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
