# Peer review of "Tungsten and Copper (II) Oxide Mixtures as Gasless Time Delay Compositions for Mining Detonators"

_materials, 2023, doi:10.3390/ma16103797_

Round 1

Reviewer 1 Report

The authors elaborate a new thermite made of W and CuO in order to fabricate gasless time-delay compositions. After presentation of the fabrication, the materials were characterized in combustion. Burn rate is ~4 – 6 mm/s. Estimated combustion temperature from heat of reaction is below 2000K. The study is very well presented and written. The authors show rigor in methodology. I just point out some minor points to address before publication :

-    1.   Can the authors better argue why they have chosen W/CuO among several thermites that burn without releasing gas. See Brotman, S et al.. A Benchmark Study of Burning Rate of Selected Thermites through an Original Gasless Theoretical Model. Appl. Sci. 2021, 11, 6553. https://doi.org/10.3390/app11146553 This paper should be cited.

-     2.  Can the authors precise the reactants size and equivalence ratio of their composition. How these two parameters may affect the burn rate and burning temperature?

Author Response

We thank the reviewer for the high evaluation of our work. Below are responses to the comments.

1. Can the authors better argue why they have chosen W/CuO among several thermites that burn without releasing gas. See Brotman, S et al.. A Benchmark Study of Burning Rate of Selected Thermites through an Original Gasless Theoretical Model. Appl. Sci. 202111, 6553. https://doi.org/10.3390/app11146553 This paper should be cited

In the introduction of the manuscript, a review of previous work on pyrotechnic mixtures with tungsten is added and justifies the choice of mixture for the study.

The components had been chosen due to the lack of information in the literature on their burn rate and mechanism of combustion. Authors have proven the gasless combustion mode and found the burn rate to be below 10 mm/s, which is a lower value than for other thermites (eg. with Al, B, Zr). That is a great advantage in terms of use in decisecond time-delay detonators.

2. Can the authors precise the reactants size and equivalence ratio of their composition. How these two parameters may affect the burn rate and burning temperature?

In the course of the research authors used micron-sized components. The D[4,3] (weighted mean value by volume) for tungsten was 10.5 microns and for copper (II) oxide was 24.3 microns. It is stated in the literature that particle size of the components affects the burn rate (the smaller grains the higher burn rate). The authors did not determine the effect of grain size on the burn rate.

Due to the formation of non-stoichiometric WOx oxides, it is difficult to determine the stoichiometry of the reaction. However, the authors determined the stoichiometric content of tungsten assuming the formation of only one oxide. This data is in line 357.

The authors do not predict a change in the adiabatic combustion temperature due to a change in the grain size of the components. In the calculations presented in chapter 3.7. sample homogeneity and complete conversion are assumed.

Reviewer 2 Report

The paper reports on the use of metallic tungsten and copper(II) oxide mixtures as the gasless time-delay compositions for detonators. The burning rates and the heats of combustion were determined for the mixtures containing different quantities of the components. The results obtained are new and worth of being published, the subject of the papers fits the scope of Materials journal.

I have the following comments:

1. In the Conclusions section, it is stated that the main combustion products of the mixture are WO2, WO2.72, WO2.96, WO3, 409 Cu2WO4, metallic copper and tungsten. Unfortunately, in the body of the manuscript, no information is provided concerning the determination of the phase composition of the combustion products. Please explain in details how WO2.72, WO2.96 and WO3 phases were discerned using the diffractograms shown.

2. The homogeneity of the starting mixtures should be analyzed thoroughly since it could affect the burning rate etc. Please provide SEM/EDX mapping data for these mixtures.

3. Please provide the data on the porosity of pressed samples (e.g. helium pycnometry data).

4. SEM/EDX mapping data for the burned mixtures could also be useful to support the conclusions made by the authors.

The quality of English is good enough.

Author Response

We thank the reviewer for his insightful evaluation of our work. Below are responses to the comments.

  1. In the Conclusions section, it is stated that the main combustion products of the mixture are WO2, WO2.72, WO2.96, WO3, 409 Cu2WO4, metallic copper and tungsten. Unfortunately, in the body of the manuscript, no information is provided concerning the determination of the phase composition of the combustion products. Please explain in details how WO2.72, WO2.96 and WO3 phases were discerned using the diffractograms shown.

Phase composition was determined using XRD library (ICDD PDF4-2020 and PDXL). Obtained diffractograms were compared with diffraction patterns of W, Cu, CuO and tungsten oxides (also non-stoichiometric oxides with defected crystal structure). Authors agree that this information should be included in the manuscript. The section 2.7. was extended with the information on the library.

2. The homogeneity of the starting mixtures should be analyzed thoroughly since it could affect the burning rate etc. Please provide SEM/EDX mapping data for these mixtures.

Indeed the homogeneity of the mixture is crucial factor. Authors used the mixing time in the turbula mixer so as to obtain the lowest possible standard deviation of the response time results. It was found that mixing over 6 hours did not change the parameters of the mixtures, hence it was assumed that this is a sufficient level of mixing for industrial applications. unfortunately, at the moment the authors do not have the possibility to perform SEM/EDX.

3. Please provide the data on the porosity of pressed samples (e.g. helium pycnometry data).

After pressing the samples inside the ZnAl time-delay element, the authors have no possibility to determine the porosity of the time delay mixture. The compressed sample is extremely compact, so gas permeation may not be possible. In addition, the use of high pressures (of the order of several hundred megapascals) may result in the formation of cavities, which may distort the result.

The porosity of the samples was calculated based on measured density and theoretical density (TMD). A corresponding paragraph has been added on page 6. The results were used to interpret the dependence of the combustion velocity on the porosity of the samples. 

4. SEM/EDX mapping data for the burned mixtures could also be useful to support the conclusions made by the authors.

The authors refer here to the answer to the second question. Unfortunately, they do not currently have the ability to perform SEM/EDX mapping of prepared samples.

Round 2

Reviewer 2 Report

The authors have addressed most of my comments. The paper is now suitable for publication in Materials.

The quality of English is good enough